# Genetic Diversity in the Portuguese Mertolenga Cattle Breed Assessed by Pedigree Analysis

**DOI:** 10.3390/ani10111990

**Published:** 2020-10-29

**Authors:** Nuno Carolino, Andreia Vitorino, Inês Carolino, José Pais, Nuno Henriques, Manuel Silveira, António Vicente

**Affiliations:** 1Instituto Nacional de Investigação Agrária e Veterinária, Fonte Boa, 2005-048 Vale de Santarém, Portugal; andreia.vitorino93@gmail.com (A.V.); ines.carolino@iniav.pt (I.C.); 2Escola Universitária Vasco da Gama, Lordemão 197, 3020-210 Coimbra, Portugal; 3Centro de Investigação Interdisciplinar em Sanidade Animal (CIISA), Faculdade de Medicina Veterinária—Universidade de Lisboa, Avenida da Universidade Técnica, 1300-477 Lisbon, Portugal; apavicente@gmail.com; 4Associação de Criadores de Bovinos Mertolengos, Rua Diana de Liz, Apartado 466, Horta do Bispo, 7006-806 Évora, Portugal; pais@mertolenga.com (J.P.); nunohenriques@mertolenga.com (N.H.); 5Ruralbit Lda, Avenida Dr. Domingos Gonçalves Sá 132, 4435-213 Rio Tinto, Portugal; msilveira@ruralbit.pt; 6Escola Superior Agrária do Instituto Politécnico de Santarém, Apartado 310, 2001-904 Santarém, Portugal

**Keywords:** native cattle, inbreeding, founders, ancestors, population structure

## Abstract

**Simple Summary:**

The conservation and maintenance of genetic diversity is one of the priorities of the Convention of Biological Diversity and is included in the United Nations (UN’s) Sustainable Development Goals. The evaluation of the genetic variability of a breed is fundamental for its future use in a sustainable way, being indispensable to outline a successful conservation or improvement strategy. Preserving genetic diversity in a population is one of the main objectives for a breed conservation program. Nevertheless, the correct management of genetic diversity is also essential for the adaptation of a population to a new environment, production system or genetic improvement. For the purpose of population monitoring, assessing changes in genetic variability and genetic erosion in animal populations, many methodologies based on pedigree analyses of inbreeding and relationships, and on the probability of genetic origin from different herds, founders and ancestors, have been used. This study presents several genetic diversity indicators in a Portuguese native cattle breed, Mertolenga, assessed by pedigree analysis, and demonstrates the usefulness of these indicators and how they can be used in the genetic management of a breed.

**Abstract:**

The Mertolenga beef cattle, currently with 27,000 breeding females in Portugal, is the largest Portuguese native breed, despite some variation in the breeding stock over the last years. The purpose of this study was to estimate parameters related to the population structure and genetic diversity and to investigate the major factors affecting genetic erosion in the breed, based on the pedigree herdbook information collected since the 1950s, including records on 221,567 animals from 425 herds. The mean generation intervals were 6.4 years for sires and 7.1 years for dams, respectively. The rate of inbreeding per year was 0.183% ± 0.020% and the correspondent effective population size was 38.83. In the reference population (35,017 calves born between 2015 and 2019), the average inbreeding and relatedness were 8.82% ± 10% and 2.05% ± 1.26%, respectively. The mean relationship among animals from the same and from different herds was 29.25% ± 9.36% and 1.87% ± 1.53%, respectively. The estimates for the effective number of founders, ancestors, founding herds and herds supplying sires were 87.9, 59.4, 21.4 and 73.5, respectively. Although the situation of the Mertolenga breed is not alarming, these results indicate the need to adopt measures to maintain the genetic variability of the population.

## 1. Introduction

The Mertolenga beef cattle breed, currently with 27,000 breeding females, is the largest of the fifteen Portuguese native breeds, although only 8000 of them are purebred. The rest are crossbred with international exotic breeds. Recently, this breed experienced a period of strong census decline caused by drastic changes in the livestock sector and unplanned crossbreeding with exotic breeds. With three quite different coat types (red, roan and red spotted; Figure 1), Mertolenga cows are used in several regions of Portugal in more than 700 farms, 216 of which are controlled by the Mertolenga’s Herdbook (MHB), mainly in the southern area of Portugal, but are also bred in other regions of the country, including the Azores Islands.

Managing the genetic diversity of a breed is fundamental for its future use in a sustainable manner in order to outline a balanced and effective strategy for conservation or genetic improvement of the population, because a limited number of breeders will inevitably lead to increased inbreeding with a reduction in additive genetic variance [1,2,3] and possibly to inbreeding depression [4]. To keep the genetic variability of a population, it is essential to take into consideration several indicators, in addition to breeding females’ total number, which allows to describe and take advantage of any genetic structure in a population [5].

Genetic characterization by pedigree analysis makes it possible to define the structure and dynamics of a population over time, considering it as a group of individuals in permanent renewal and taking into account its gene pool [6]. The use of pedigree analysis for measuring genetic diversity is referred by several authors and in many species [7,8,9,10] as a great methodology to study populations, as it describes the genetic structure and its evolution through generations. It also contributes substantially to animal breeding and conservation genetics research, resulting in many applications [5,6,7,8,9,10,11,12,13,14,15,16,17,18].

The main goal of this study is to describe genetic diversity in the Mertolenga cattle breed based on pedigree analysis, taking into consideration their different coat types, and identifying factors that may have affected the population, in order to establish conservation strategies and minimize further losses in terms of genetic diversity.

## 2. Materials and Methods

### 2.1. Ethics Statement

The Animal Care and Use Committee’s approval was not obtained for this study because the data was gathered from the Mertolenga Breeders Association (ACBM) digital herdbook database.

### 2.2. Animals

The Mertolenga breed is a medium sized animal with a red or reddish coat with white patches and white hook-shaped horns. It is known for its remarkable maternal features and adaptability to poor feeding conditions, being traditionally raised in extensive environments, on farms, normally with reduced densities that have, on average, 70 breeding females, but that can range from 5 to 600 cows per herd.

In the 1970s, this population was under serious threat of extinction due to land restructuring of the Portuguese livestock and uncontrolled crossbreeding with exotic breeds. However, in 1978, the Mertolenga’s Herdbook (MHB) was created and the population grew/increased during the following decades. Currently, there are around 27,000 breeding females, of which 14,500 cows are registered in the herdbook under the Breeders Association administration, and only 8000 are used for pure breed reproduction.

Depending on the Portuguese region, the environmental conditions, the soils and the beginning (and consequently the duration) of the grazing seasons, births are recorded throughout the year but with a peak in the winter–spring period. Some farms use defined mating seasons (3 to 6 months), while others keep the bull in the herd all year long. Reproduction takes place essentially by natural mating, with very limited use of artificial insemination.

Breeders usually produce their own replacement females, while bulls are either from their own herd or acquired from other herds. From 2003 onwards, a genetic evaluation by Best Linear Unbiased Prediction (BLUP)–Animal Model methodology was implemented and applied to reproductive and maternal traits [19]. Since 2010, only animals with parentage testing confirmation by microsatellite markers and accepted morphological evaluation can be registered as breeders in the breeding section of MHB.

### 2.3. Data

All the information used in this study was obtained from MHB records and is available through the online database Genpro [20]. These records contain information referring to individual identification numbers, gender, birth and cull date, sire and dam identification, herd of origin and current herd. Data analyzed included records on 209,503 animals registered in the herdbook between 1978 and 2019, together with additional pedigrees from 10,064 ancestors born between the 1950s and 1978, for a total of 221,567 animals from 425 herds.

The dataset information was used to assess the number of registered animals (dams, sires and calves) and herd size over the years, age distribution of sires and dams, number of offspring per progenitor and several useful demographic and genetic parameters to study the evolution of the genetic variability of the Mertolenga population.

### 2.4. Pedigree Analysis

Demographic analyses were performed with software specifically developed for this purpose [5] and were indicated with the ENDOG v4.8 software [21].

The equivalent number of complete generations known per animal (*n_i_*) was used to assess the degree of pedigree completeness, and calculated as:(1)$$n_{i}\;=\;\frac{n_{s}\;+\;1}{2}+\frac{n_{d}\;+\;1}{2}$$
where *n_s_* and *n_d_* are the number of generations known for the sire and dam respectively, when *s* and *d* are known; if *s* or *d* are unknown, then *n_s_* or *n_d_* respectively, assume the value of 1. Base animals were assigned a number of generations known equal to 0.

Individual coefficients of inbreeding (*F_x_*) and additive genetic relationships among pairs of animals (*a_ij_*) were computed based on the numerator relationship matrix among all animals [22], with software developed by Carolino and Gama [5]. The estimates of *F_x_* and *a_ij_* were used to evaluate the average kinship relationship between animals born on the same, or different farms, and the average animal inbreeding.

The regression coefficient of individual inbreeding on year of birth was obtained with the General Linear Model (GLM) procedure of SAS^®^ 9.4 [23], and this was considered to be the rate of inbreeding per year (∆*F*/year).

Generation intervals were calculated for sires and dams of all born calves, and for the 4 paths of selection (average age of sires of sires, sires of dams, dams of sires and dams of dams). The generation intervals for the 4 paths of selection were averaged to obtain a pooled generation interval (*L*), which was used to compute the rate of inbreeding per generation (Δ*F/g*) as Δ*F/g = L* (Δ*F/year*). In addition, the effective population size (*N_e_*) was then calculated as described by Falconer and MacKay [1]:(2)$$N_{e}=\frac{1}{2\left(\frac{\Delta F}{g}\right)}$$

The genetic contributions to the reference population of founder animals, ancestors and founding herds were computed, as described by James [12], Lacy [13] and Boichard et al. [14]. The effective number of founders (*f_e_*), ancestors (*f_a_*) and herds (*f_h_*) were computed from those genetic contributions as:(3)$$f_{j}=\frac{1}{\sum_{k=1}^{n}q_{k}^{2}}\;$$
where subscript *j* stands for founders, ancestors or herds, depending on the situation considered, and *q_k_* represents the genetic contribution of a given founder or founding herd, or the marginal contribution of an ancestor [14].

The founder genome equivalents (*f_g_*), defined as the number of founders that would be expected to produce the same genetic diversity as in the population under study if the founders were equally represented and no loss of alleles occurred [13], was obtained by the inverse of twice the average co-ancestry of the individuals included in a pre-defined reference population, following Caballero and Toro [16].

The average relatedness (*AR*), which corresponds to the mean relationship of each individual with all animals in the pedigree, was computed for each individual, with ENDOG v4.8 [21]. This software was also used to compute the genetic contributions of different herds supplying sires, paternal grandsires and great-grandsires to the reference population, according to Robertson [11].

For the purpose of this study, several reference populations were considered defined by sets of animals born in certain periods of time (per year or per 5-year periods).

## 3. Results

Annual evolution of the number of registered cows (all breeding females and purebred breeding females), bulls, calves and herds in the MHB, for the period from 1990 to 2019, is presented in Figure 2. Currently, there are approximately 27,000 registered cows, including nearly 8000 cows used in purebred mating, 190 bulls and 216 herds enrolled, where about an average of 6500 pure calves were born/year over the last years.

Overall, the total number of breeding females increased until 2014, but has decreased since then, and the number of purebred cows has been gradually decreasing since 2017, on average −280 cows each year.

The number of active breeding bulls (with offspring) reached a maximum in 2007 (274 males) and subsequently decreased to values that have ranged between 189 and 240 bulls depending on the year (Figure 2), following the same tendency of decline as the use of cows in the pure line. In 2019, there were 189 active Mertolenga breeding bulls registered in the MHB.

The number of pure calves registered annually in the birth section of the MHB reached its highest value in 2007 (11,149 calves). Since then, the trend has been to decrease, although with some oscillations, registering since 2010 values between 6300 and 7500 calves per year, which corresponds to an average of 6900 births of purebred Mertolenga animals per year.

The maximum number of breeders with Mertolenga females was observed in 2009 (277), but the number of breeders where purebred animals are born has been decreasing since 2007, when the maximum was reached (205 breeders). In 2019, Mertolenga calves were born in only 138 breeders (64% of total breeders) registered in herdbook.

In the last 5 years (2015–2019), the average herd size was 103.5 ± 127.44 registered cows with 27.2 ± 53.5 pure calves per herd/year, with more than 50% of the herds registering less than 10 purebred calves each year and about 40% not registering any Mertolenga calves.

The evolution of the number of breeding females registered in the MHB according to each coat variety was slightly different (Figure 3). Although red spotted cows continue to exist in smaller numbers, proportionally and in value, they have been increasing, just like roan females, in contrast to the red variety, which has been decreasing. Regarding calves, in addition to observing a decrease in total births, a more pronounced decrease in red animal births is noted. In recent years, the proportion of existing breeding females is 13% red spotted, 54% roan and 33% red, whereas births correspond to 18% red spotted, 53% roan and 29% red.

The age distribution of sires and dams of 221,567 calves born between 1978 and 2019 is shown in Figure 4. The age distribution of females at birth is according to the bovine species and confirms the capacity of the Mertolenga cows to remain active in production until an advanced age. More than 30% of the females give birth beyond the age of 10 years and more than 6% of the females keep breeding beyond the age of 15 years.

Most breeding males (53%) are between 4 and 8 years old when their offspring is born (Figure 4). A small percentage of males (5%) are parents after the age of 12, with a slight increase in the percentage of bulls being parents at an older age.

The mean ages for sires and dams at calves’ birth were 6.4 ± 2.6 and 7.1 ± 3.7 years respectively, with the first offspring usually produced at around 3 years of age in cows (34.8 ± 7.5 months), and somewhat later in bulls. These results indicate that, when compared with cows, the use of bulls started at a later age (37.3 ± 9.95 months), but that they were also replaced at a faster rate. Nevertheless, nearly 10% of the offspring was sired by bulls at, or above, 10 years of age, whereas the percentage of calves produced by cows of the same age range was about 24%. In recent years, the percentage of females that calved at 10 or more years of age was about 36% (results not shown). The results obtained for the generation intervals (*L*) (Table 1) for calves had an overall mean of 7.05 years and were slightly higher for cows than for bulls. Similarly, when generation intervals were evaluated for the 4 paths of selection, they were almost 1 year longer for the dam paths.

The 204,671 calves with a known sire and born between 1978 and 2019 were the offspring of 1629 sires, with an average of 133.7 ± 141.4 calves/sire.

The distribution of the number of breeding males according to the number of offspring presented in Figure 5 shows great differences between the numbers of calves per sire. More than 60% of the animals born are sons of only 20% of the bulls, and 11% of the animals born are sons of only 2% of the bulls (36 animals). In the opposite direction, more than 12% of the bulls have less than 25 offspring throughout their lives, representing only 1% of the births.

The degree of pedigree completeness available for the first 3 generations is illustrated in Figure 6a,b for all calves and for those born in the period from 2015 to 2019 (reference population). The degree of pedigree completeness has clearly evolved through the years since the beginning of the herdbook and calves born between 2015 and 2019 have more than 95% of the grandparents and great-grandparents known (Figure 6c). Almost every animal born between 2010 and 2014 have parents, grandparents and great-grandparents known, although in the following period, from 2015 to 2019, a small increase of animals born without a known father was observed (Figure 6c). Even so, in the last 5 years, it is observed that when a bull is used as a breeder, its pedigree via parental to the grandparents is known.

The degree of pedigree completeness was also estimated by the number of generation equivalents, which corresponds to the average number of complete generations known in the pedigree of an individual. The number of generation equivalents increased from nearly 2 for calves born in the 1990s to close to 6 generations for calves born in 2019 (Figure 7a).

The evolution of inbreeding (Figure 7a) indicates an irregular increase over time, from a mean value of 3.4% for calves born in 1990, to an average of 9.3% for calves born in 2019. Inbreeding exceeded 9% in 2003. In the following years, it decreased and remained between 7% and 9% until 2016, but in 2017, it was again above 9%. The percentage of inbred animals born has also been increasing sharply over the years (Figure 7a), remaining above 85% since 2013. The average relatedness (*AR*) increased from 1990 to 2009, reaching 2.12% in 2009, and since then has remained between 2.0% and 2.2%.

Although the proportion of animals born from the 3 coat types (red spotted, roan and red) is quite distinct over the years, until 2015, the evolution of inbreeding was very similar. However, this evolution was very different in the red coat from 2016, reaching an average inbreeding of more than 13% in 2019, while red spotted and roan animals, on average, are born with less than 9% inbreeding (Figure 7b). In the reference population, the mean inbreeding per herd ranged between 0.0% and 30%, as shown in Figure 8. Most farms (52%) have mean inbreeding values between 5% and 10%. In almost 20% of the herds, animals are born with more than 15% inbreeding on average and only one farm was observed where animals born during this period were non-inbred.

The evolution of kinship between animals born in the same herd and between animals born in different herds, regarding inbreeding over 10 years, is illustrated in Figure 9 and shows some variability between years in the average kinship between animals born in the same herd (ranging from 27.1% to 34.6%). In 2017, the highest value was observed. The average kinship values between animals born in different herds are low (<2.1%) and with less variation between years (from 0.95% in 2015 to 2.15% in 2018).

The average relationship in the reference population and for animals from the same herd was 29.25% ± 9.36% and for animals born in different herds was 1.87% ± 1.53% (Table 2). The average relatedness in the whole population was 1.76% ± 1.55%, and in the reference population was 2.05% ± 1.26% for animals born from 2015 to 2019.

The linear regression of inbreeding per year of birth resulted in an estimated rate of inbreeding of 0.183% ± 0.020%/year (*p* < 0.01). From the annual rate of inbreeding and the mean generation interval (7.05 years), the estimated rate of inbreeding per generation was 1.29%, and the corresponding effective population size was 38.83 (Table 2). The estimated effective population size according to Gutierrez et al. [17], via regression on equivalent generations for a given subpopulation, was 41.69.

The results of the retrospective evaluation of cumulative genetic contributions of founders, ancestors and herds to the reference population are summarized in Table 3 and represented in Figure 10. Overall, there were 11,471 founders, of which 337 were sires and 11,134 were dams.

Considering the births from the last 5 years (2015–2019), the effective number of founders, ancestors and founding herds were, respectively, 87.9, 59.4 and 21.4 (Table 3).

The cumulative genetic contributions of the most important founders, ancestors and herds (Figure 9) show an abrupt increment in the early stages of the curves, indicating that a small number of animals and herds have a strong influence on the genetic variability of the breed.

Overall, 50% of the genetic pool of the reference population is accounted by the contribution of 56 founders, 41 ancestors and 9 herds (Table 3), with the 5 most influential founders, ancestors and founding herds contributing, respectively, 19.6%, 22.1% and 35.2%.

The estimate of founder genome equivalents (*f_g_*) to the reference population was 48.7. This estimate indicates that, from the 11,471 founders of the population, if they all were equally represented and no loss of alleles occurred, only 48.7 would produce the genetic diversity currently existent.

The number of founders, ancestors and herds over 4 periods of 5 years (Figure 11) shows a decreasing trend and, consequently, some genetic bottleneck in the population. The estimates of these three indicators from 2015 to 2019 were approximately 17% lower than estimates for the period from 2000 to 2004. The higher the *fe/fa* ratio, the higher the existence of bottlenecks along the pedigree [14]. The *fe/fa* ratio has changed between 1.40 and 1.53, and reached its highest value from 2005 to 2009. The various genetic contribution indicators suggest a rather unequal contribution of founders, ancestors and herds to the current gene pool and to the existing population at different times.

## 4. Discussion

Despite its good fertility and remarkable rusticity, combined with a regular maternal capacity [24], with excellent results in terminal crossings as maternal line, the Mertolenga cattle breed suffered a great decrease in the mid 1970s. Notwithstanding a population of 25,000 adult breeders in 1975, with the decrease already underway, this number declined to around 2000 animals in the early 1980s.

Several factors prevented the extinction of the Mertolenga cattle breed and reversed the trend in the census size, including national policy strategies, establishment of the herdbook, creation of a performance test station in the Baixo-Alentejo Experimental Center (CEBA)—Abóbada Farm, in 1978, recognition of a protected designation of origin (PDO) approved by the European Union in 1994, the inclusion of Mertolenga in the group of breeds at risk and eligible for agri-environment programs that provide financial support in the framework of the European program for conservation of genetic resources, and several breeders’ support activities promoted by Mertolenga Breeders Association (ACBM). All these strategies and activities combined contributed to the resurgence of the Mertolenga breed from a limited base. It currently represents the biggest native beef cattle breed in Portugal, both as a pure breed and mainly as the maternal line in organized crossbreeding programs.

Due to the maternal characteristics of the breed and their use as a maternal line, more recently, ACBM started a program to promote F1 females (exotic bull × Mertolenga female) as a maternal line “Mertolenga F1 Program”, in which the main exotic breeds used in Portugal are collaborating (Limousine, Aberdeen-Angus, Charolais and Salers).

Currently, there are approximately 27,000 breeding females (Figure 2). Even so, only about 14,500 are registered as breeding stock in the MHB and only 8000 of them are purebred. Furthermore, with a decreasing trend in the last five years of breeding females, cows in the pure line, calves, bulls and herds, the Mertolenga breed requires some attention. The breed’s recovery from a narrow base, and a strong influence of very few herds and sires, particularly animals from a herd belonging to the Portuguese Ministry of Agriculture in CEBA, can bring unavoidable consequences for the genetic breed structure.

In the beginning of the 21st century, the Mertolenga cattle breed was classified as “non-threatened” according to the criteria used by the European Union (EC Regulation No. 45/2002) to define the animal genetic resources risk status. Recently, in 2015, according to the document of delegated acts for the new Rural Development Regulation adopted by the European Commission, Mertolenga is considered internally to be at risk of extinction, with grade C (lower risk).

The different percentages of animals in terms of coat type (red spotted, roan and red; Figure 3) are related to some degree to tradition in the various farming areas, but more to breeders’ preferences, resulting from the valorization of calves. In general, at weaning, red spotted and red animals have less commercial value.

Longevity is a desirable attribute of the Mertolenga, but it has an impact on breeding programs of long generation intervals in the dam paths of selection. The vast majority of Portuguese native cattle breeds, Mertolenga in particular, have the capacity to remain in production until older ages (Figure 4). Between 2015 and 2019, cows had an average calving age of 7.70 ± 3.99 years. Additionally, bulls are used in reproduction for a long period of time (mean age at calves’ birth of 6.40 ± 2.61 years), with a subsequent impact on the observed rate of inbreeding.

Although artificial insemination is seldomly used, a small number of famous bulls have a very large number of offspring by natural mating: 36 bulls (2%) have more than 500 sons each and are responsible for more than 11% of the births (Figure 5), which also contributes to a high inbreeding rate. This imbalance in the number of offspring per sire usually occurs in local breeds with a low population size, but it also occur in larger ones, when included in classic selection schemes, as is the case with dairy cattle [25], where several factors allowed the selection of a few groups of bulls each year, each one having a very large number of offspring.

To study the genetic variability of a population and the main factors of genetic erosion, it is essential to carry out a broad study over time, taking into account several generations, both in terms of individuals and farms. It is essential to know several indicators and to evaluate them dynamically over the years, such as the size of farms, the age structure of the herd, bottlenecks in terms of excessive use of some breeders and their genetic contribution, the exchange of genetic material between herds, among others.

Controlling the rate of inbreeding is usually one of the major targets in conservation and selection programs [16,17,18,19,20,21,22,23,24,25,26,27,28,29,30,31], and more so when selection decisions take family or genomic information into account, such as in BLUP, GBLUP (Genomic best linear unbiased prediction) and ssGBLUP (Single-step genomic BLUP) [32,33,34,35,36], or when reproductive biotechnology (Embryo transfer–ET, in-vitro fertilization–IVF and Multiple Ovulation Embryo Transfer–MOET) is used [37]. The evaluation of genetic diversity of populations has evolved over the years, with the advantage that we can now benefit from methodologies, criteria and concepts proposed longer ago and those implemented more recently. According to Carolino and Gama [5] and Vicente et al. [7], parameters based on the probability of genetic origin from different herds [11], founders [12,13] and ancestors [14] have also been used to assess changes that occurred in the population over a short period of time [14]. These principles have been applied to the genetic characterization of different cattle breeds, including some Portuguese native cattle breeds, such as Alentejana [5,6,7,8,9,10,11,12,13,14,15,16,17,18,19,20,21,22,23,24,25,26,27,28,29,30,31,32,33,34,35,36,37,38], Barrosã [39], Preta [40], Jarmelista [41], Marinhoa [42], Arouquesa [43] and Cachena [44], and from other countries, such as Brazilian Marchigiana [45], Bonsmara [45], Brahman [46], Mexican Simmental [47], Spanish Alistana, Sayaguesa, Avileña—Negra Ibérica, Morucha, Asturiana de los Valles, Asturiana de la Montaña, Pirenaica, Bruna dels Pirineus [48,49], Lidia [50], French Charolais and Limousin [51] and Japanese Brown [52] cattle.

In Table 4, a summary of the main demographic parameters reported in the literature for several cattle breeds is presented, covering a wide range of census numbers and management systems. Although there has been a slight decrease in the number of known parents in animals born between 2015 and 2019, the level of pedigree completeness in Mertolenga is currently good (Figure 6a,b), especially when compared to other breeds kept in similar extensive production systems in Portugal [38,39,40,41,42,43,44] or in Spain [48]. The slight decrease observed in the number of known parents (bulls) between 2015 and 2019 was due to the amendment of an internal regulation that now allows the registration of calves on the birth side section of MHB, based on a declaration of mating. Thus, the percentage of sires known went from 99.6% in 2010–2014 (results not shown), to 96.5% in 2015–2019, resulting in a decrease of known parents by 1.5% (Figure 6b). However, the requirement to do parentage testing confirmation by microsatellite markers was maintained to register all candidate animals as breeders in the breeding section of the MHB.

The complete known generations (*n_i_*) increased, on average, by 0.142 ± 0.002 years (results not shown) since 1990 and is presently nearly 6 generations for animals born in 2019. This value is higher than the one for some breeds whose herdbook started at the same time, and close to the values obtained in the Charolais and Limousine breed reported by Bouquet et al. [51]. It is higher than that in several native Spanish breeds [48,49,50], but lower than some international breeds with a longer established herdbook (e.g., Charolais and Limousine) that have a *n_i_* above 15 [57].

The average inbreeding per year of birth (Figure 7a) showed a growing trend, although not constant, with a decrease in 2005, as a result of greater attention by ACBM, an efficient advisory service for breeders in defining matings, and the beginning of a red spotted herd recovery plan, particularly evident in Figure 7b. This plan allowed to diversify the use of sires and fostered targeted pairings in order to minimize the kinship between breeders. The increase in inbreeding over the years is due, on one hand, to the mating of increasingly related animals (a result of the real increase in the average kinship between breeders), and, on the other hand, to the increase in pedigree information that provides a more accurate estimate of the value of inbreeding coefficients.

Boichard et al. [14] reported that the estimation of the individual inbreeding coefficient is very sensitive to the quality and quantity of available pedigree information. The average inbreeding of 9.3% observed in Mertolenga calves born in 2019 is higher when compared to other beef cattle breeds, but below the average value of 14.4% stated by Burrow [4] for several beef breeds, in his wide analysis about the effects of inbreeding in beef cattle. The estimate of inbreeding is between an average of 8.5% in Alentejana calves born in 2003 [5], an average of 9.6% in Alentejana calves born in 2016 [38] and additional estimates obtained in other Portuguese native breeds such as Barrosã (5.2%) [39], Preta (6.0%) [40], Jarmelista (14.1%) [41], Marinhoa (4.5%) [42] and Cachena (6%) [43]. The high percentage of inbred animals in Mertolenga’s reference population (86%), added to a considerable percentage of animals (17%) that have coefficients of inbreeding above 25.0% (results not shown), indicates that there is a need for some measures to be taken to slow the rate of inbreeding in the Mertolenga population.

In the reference population, the mean inbreeding per herd ranged between 0.0% and 30%, as shown in Figure 8, being on average 7.93% ± 5.54%. Most farms (52%) have mean inbreeding values of between 5% and 10%. In almost 20% of the herds, animals are born with more than 15% inbreeding on average, and only one farm was observed where animals born during this period were non-inbred. These results demonstrate the different options of breeders when it comes to inbreeding and its consequences. In some cases, particularly in red and red spotted animals, there are greater difficulties in conducting male exchanges between farms and defining pairings that minimize kinship.

These results are related to the evolution of kinship between animals born in the same herd and between animals born in different herds, as illustrated in Figure 9. It shows some variability between years and that average kinship values between animals born in different herds are low (<2.1%).

The average kinship between animals born on the same farm and year is naturally higher than the average kinship between animals born on different farms. The results obtained (Figure 8 and Figure 9) still seem to show that, in some years, breeders had more concern with minimizing the kinship between animals.

The annual rate of inbreeding (Δ*F* = 0.183%/year) estimated for the entire Mertolenga population enrolled in the MHB (209,503 animals born between 1978 and 2019) is lower than that obtained for the Alentejana breed for animals born between 1968 and 2003 (0.33%/year) [5], in animals born between 1968 and 2016 (0.23%/year) [38], and lower than the rates found in the Morucha, Asturiana de la Montaña and Alistana [48,49] Spanish breeds (Table 4). This estimate (Δ*F* = 0.183%/year) is inferior to those obtained for the Portuguese breeds Arouquesa [43] and Preta [40] and in the Italian Sardo Bruna and Sardo Modicana [57]. The annual rate of inbreeding estimated for Charolais and Limousine (between 0% and 0.03%/year) is also lower than the ones for Mertolenga. When the population or period under study includes animals with little pedigree information (*n_i_* < 2), naturally, these will have zero or very low levels of inbreeding, which together with individuals born more recently and with more pedigree information, may overestimate the previous estimate of annual rate of inbreeding.

The estimate of the effective population size obtained in the Mertolenga breed (*N_e_* = 38.83) is lower than the one recommended (*N_e_* = 50) as the minimum number to maintain genetic diversity, both in conservation [26,27,28,29,30,31,32,33,34,35,36,37,38,39,40,41,42,43,44,45,46,47,48,49,50,51,52,53,54,55,56,57,58,59,60,61,62,63] and in selection programs [64], and is a warning about the need to reassess the management of the breed in the future. This estimate is in the range of other Portuguese native breeds, such as Alentejana, Barrosã and Cachena, and exotics such as Japanese Black, Mucca Pisana, Sayaguesa and Morucha (Table 4).

The size of the *N_e_* between a local livestock breed, in many cases < 50, and an international breed of worldwide dimension (e.g., Charolais, Limousine or Blonde d’Aquitaine), in some cases with *N_e_* > 500, is very uneven. These are distinct realities and also very different possibilities of managing the existing genetic variability. Inbreeding rate per generation should be below 1.0% [26,27,28,29,30,31,32,33,34,35,36,37,38,39,40,41,42,43,44,45,46,47,48,49,50,51,52,53,54,55,56,57,58,59,60,61,62,63,64,65], which may preclude selection when the population size is particularly small. However, as the population size increases, selection intensity can be increased, resulting in a continuum of situations with respect to selection differential.

Of the 11,471 founders who constituted Mertolenga cattle breed, large differences were observed in their genetic contributions to the current population, with five founders contributing to about 19% of the actual genetic pool. The effective number of founders (*f_e_*) was 87.9 between 2015 and 2019, and has been decreasing since 2000–2004, when it was 104.82. The slight increase of the *f_e_* between 2015 and 2019 compared to the previous period is due to the fact that in the last 5 years, there have been some registered animals in the MHB (side section) without confirmed paternity, as explained above. The effective number of ancestors (*f_a_*) is currently 59.4 and also shows a decreasing trend during the periods studied. However, these values are not as low as the ones observed in other breeds, including some Portuguese ones, summarized in Table 4, where the lowest values are observed for Jarmelista (Portugal), Mucca Pisana (Italy) and Lidia (Spain).

The *f_e_*/*f_a_* ratio in Mertolenga is currently about 1.5, having ranged from 1.40 to 1.53, indicating that some bottlenecks have occurred in the pedigrees, as expected from some influence of a few prominent sires and herds (Figure 10). This ratio is dramatic in some breeds, such as Sayaguesa, Alistana, Asturiana de los Valles and Charolais (Table 4). According to Bouquet et al. [51], the *fe*/*fa* ratio in the Charolais breed has been increasing since the early 1990s, indicating that bottlenecks have lasted two decades. In beef cattle breeds with large dimensions, some results derived from probabilities of gene origin, indicating that the genetic diversity within each subpopulation is still relatively large, especially when compared with dairy cattle populations. However, it follows a slight declining trend, as already suggested by the analysis of inbreeding trends.

The strong influence of a reduced number of Mertolenga herds is revealed by the fact that 5 herds alone account for about 40% of the actual gene pool, and the effective number of herds supplying sires, grandsires and great-grandsires is, respectively 73.5, 33.5 and 17.8. These values are higher than the ones observed in the Alentejana cattle breed but suggest that a selection nucleus was developed in each breed, and that most herds have been influenced by the selection practiced in each of those nuclei.

The relationships between *N_e_*, *f_e_* and *f_a_* provide information about bottleneck occurrence in one population, such that *f_e_* should be close to *N_e_*/2 in a population where genetic drift has stabilized [16,17,18,19,20,21,22,23,24,25,26,27,28,29,30,31,32,33,34,35,36,37,38,39,40,41,42,43,44,45,46,47,48,49,50,51,52,53,54,55,56,57,58,59,60,61,62,63,64,65,66] and the *f_e_*/*f_a_* ratio should be close to 1 if important bottlenecks have not occurred in the population [14].

The rate and level of inbreeding observed in Mertolenga are not, in fact, alarming and are in accordance with the values observed in other cattle breeds (Table 4). These trends show that Mertolenga has some genetic erosion, and that taking steps to control inbreeding is justified.

Several studies have presented strategies to control and minimize the effects of inbreeding in conservation programs, some of which include maximization of genetic contributions from different ancestors, with restrictions on the distribution of parents over age class or minimization of co-ancestry selection for overlapping generations. Others, for example, offer recommendations on effective population size and generation intervals [63,64,65,66,67,68,69,70,71,72].

Various methods of controlling inbreeding in selection programs have been suggested, including strategies to achieve genetic response with control of inbreeding, ranging from the earlier methods using sub-optimal criteria of selection, creation of sublines, restrictions on family size, restrictions on BLUP application, considering genetic relationships among selected animals, optimized mating programs and appropriately weighting breeding value estimates, to inbreeding generated by selection decisions [26,27,28,29,30,31,32,33,34,35,36,37,38,39,40,41,42,43,44,45,46,47,48,49,50,51,52,53,54,55,56,57,58,59,60,61,62,63,64,65,66,67,68,69,70,71,72,73,74,75,76,77,78,79,80].

## 5. Conclusions

Nowadays, conservation of genetic diversity is universally accepted as being essential for the sustainable use of animal genetic resources and therefore to the future of humanity.

Pedigree analysis was useful when monitoring changes in the population structure and gathering important demographic parameters in the Mertolenga population. The average inbreeding coefficient per year of birth in Mertolenga cattle has increased over time, although current estimates of inbreeding rate and effective population size are not in the range of critical levels. Since the beginning of the Mertolenga’s Herdbook, an excessive use by some breeders has been observed, also confirmed through the genetic contribution of founders, ancestors and herds to the existing population at different times. The fact that in this type of study, a vast set of demographic indicators can be analyzed, makes it possible to diagnose the changes in the genetic structure and the main factors that have influenced it.

This work also shows that different indicators of variability can be used to assess the risk of extinction of a population, and not just one mainly used, such as the number of adult breeding females. In real populations, the conditions of an ideal population are not found, so a single evaluation criteria to assess the risk of extinction of a population may not be appropriate in different situations. The estimate of the average degree of kinship between animals born in different herds compared to the estimate of animals born in the same herd seems to indicate that, currently, there are conditions, through a correct demographic management of the Mertolenga bovine breed (globally and to each of the 3 coat types), for the values of the individual inbreeding to remain acceptable without major damage.

The pedigree information available in the herdbook is quite complete, but the control of affiliation by DNA analysis should be maintained. It seems that current mating practices to avoid inbreeding work well when used. For long-term maintenance of genetic diversity and dynamics of the breed, the minimization of genetic relationships between candidate breeding animals is the most promising approach, moreover, with the aid of knowledge from molecular genetics.

Measures should be taken to avoid further losses in genetic variability in the Mertolenga breed, including the selection of breeding animals with a more extensive representation of ancestors and with a lower relationship with the population, and the rotation of animals among herds, particularly among herds of the same coat type. Restrictions may be applied on BLUP-selection, based on the expected impact of inbreeding on the herd where the animal will be used. Breed management should include in situ or ex situ conservation actions, with conservation nuclei of the different coat types.

## Figures and Tables

**Figure 1 animals-10-01990-f001:**
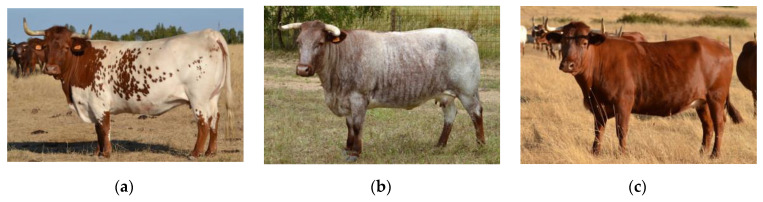
Different Mertolenga coat types: (**a**) red spotted, (**b**) roan, (**c**) red.

**Figure 2 animals-10-01990-f002:**
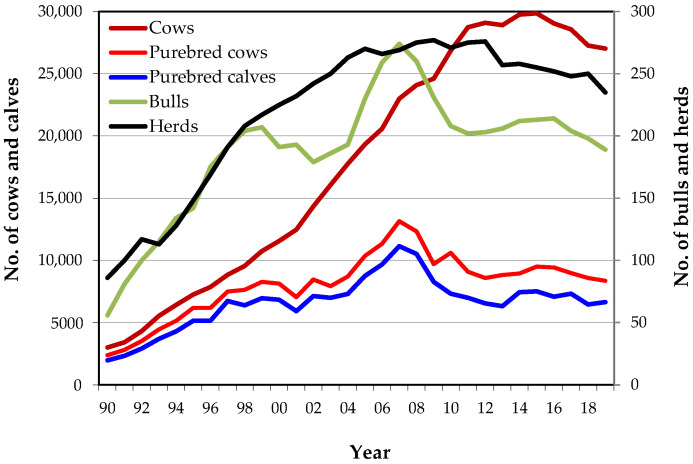
Number of cows, bulls, calves and herds registered in the Mertolenga’s Herdbook by year.

**Figure 3 animals-10-01990-f003:**
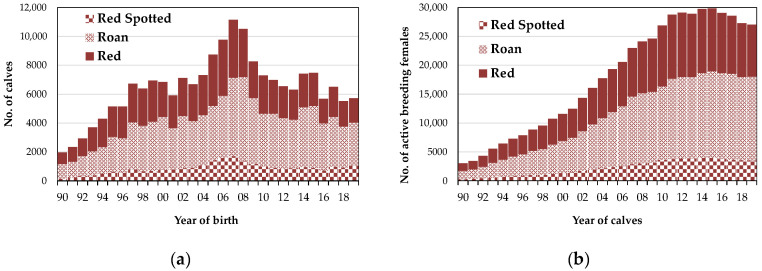
Number of animals registered in the Mertolenga’s Herdbook by year and coat variety. (**a**) Calves, (**b**) breeding females.

**Figure 4 animals-10-01990-f004:**
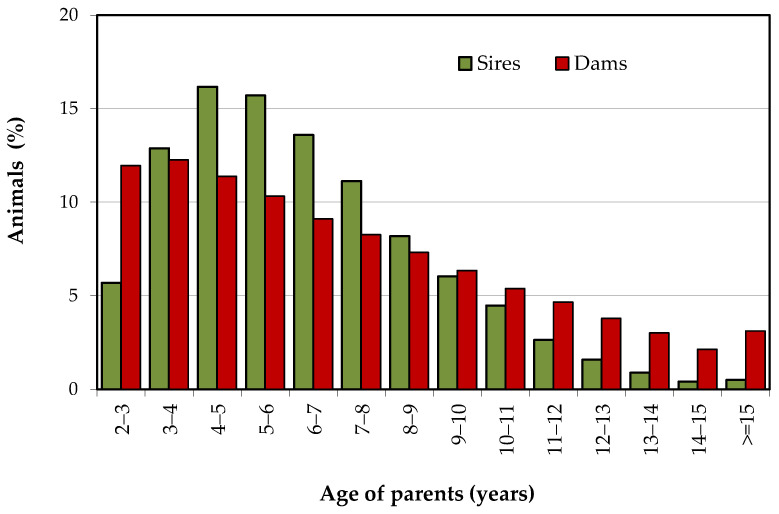
Age distribution of parents of calves in the whole population (calves born between 1978 and 2019).

**Figure 5 animals-10-01990-f005:**
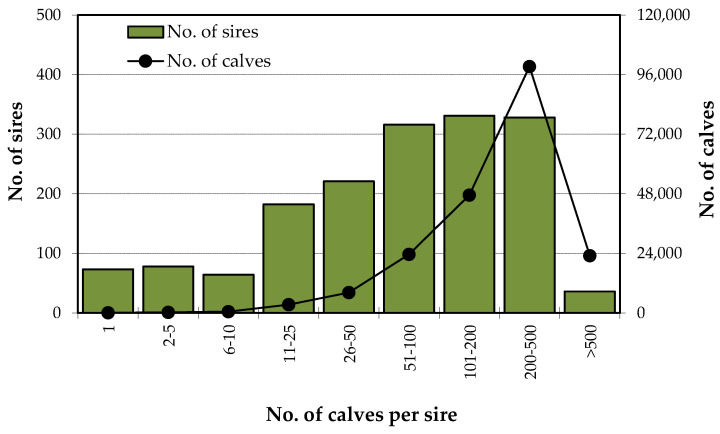
Number of sires and calves, by classes of number of calves per sire.

**Figure 6 animals-10-01990-f006:**
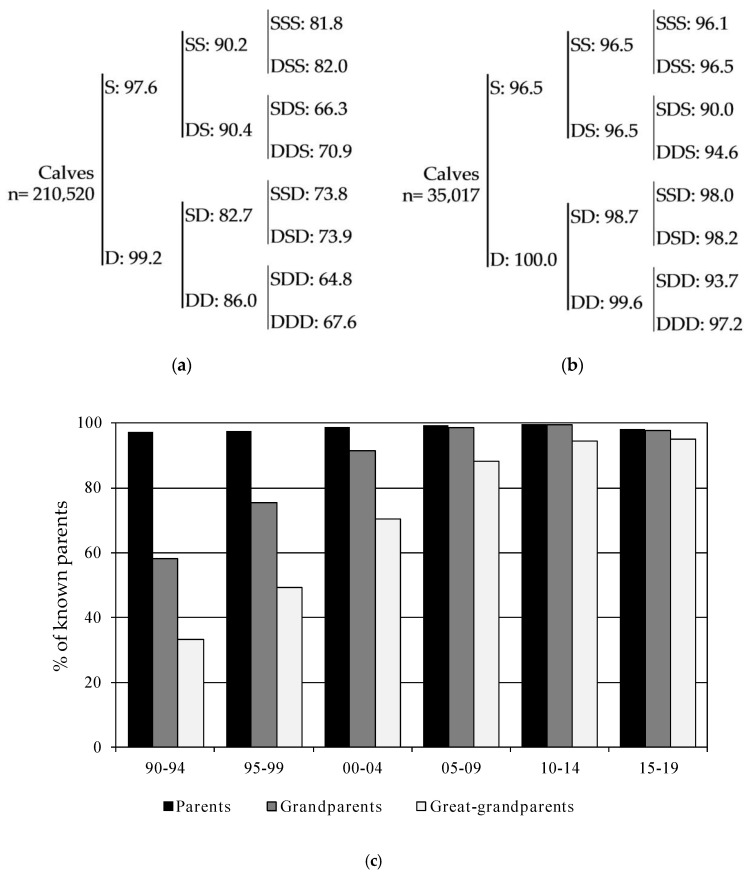
Average percentage of sires (S), dams (D), paternal and maternal grandparents (SS, DS, SD, and DD), and great-grandparents (SSS, DSS, SDS, DDS, SSD, DSD, SDD, and DDD) known for: (**a**) The whole population, (**b**) calves born between 2015 and 2019 and (**c**) average percentage of ancestors known by periods of 5 years.

**Figure 7 animals-10-01990-f007:**
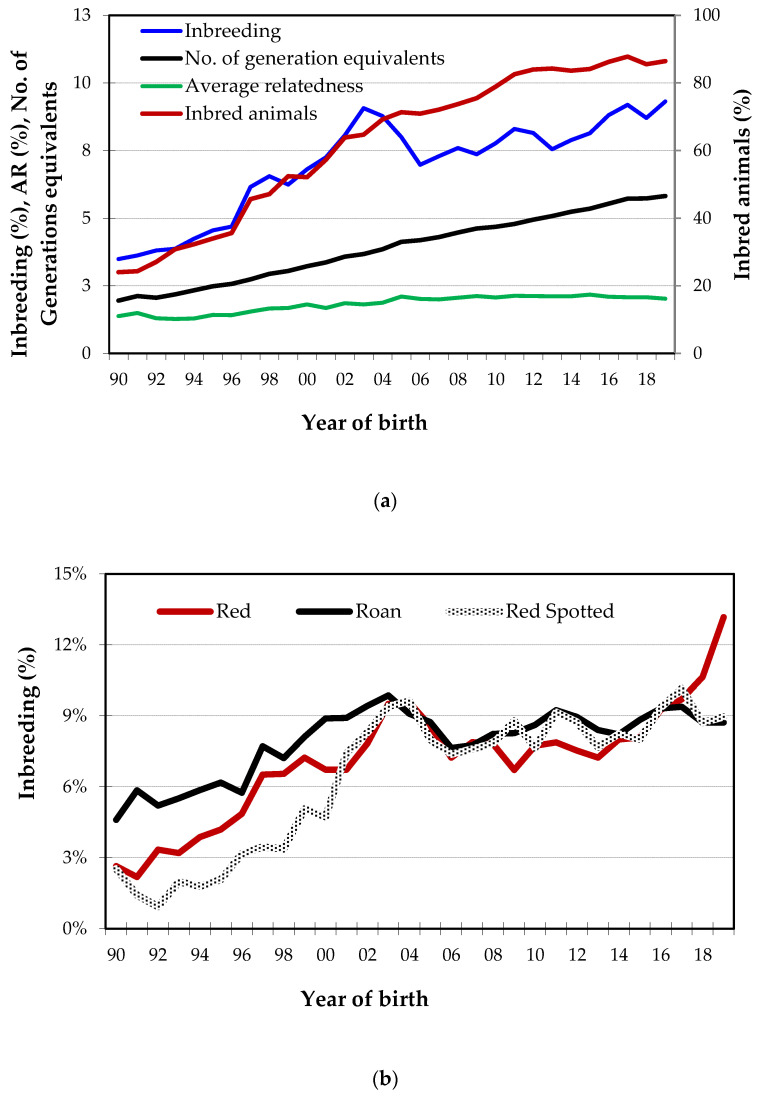
Average inbreeding per year of birth. (**a**) Average inbreeding, number of generations known, average relatedness and percentage of inbred animals. (**b**) Average inbreeding by variety (coat color) and year of birth.

**Figure 8 animals-10-01990-f008:**
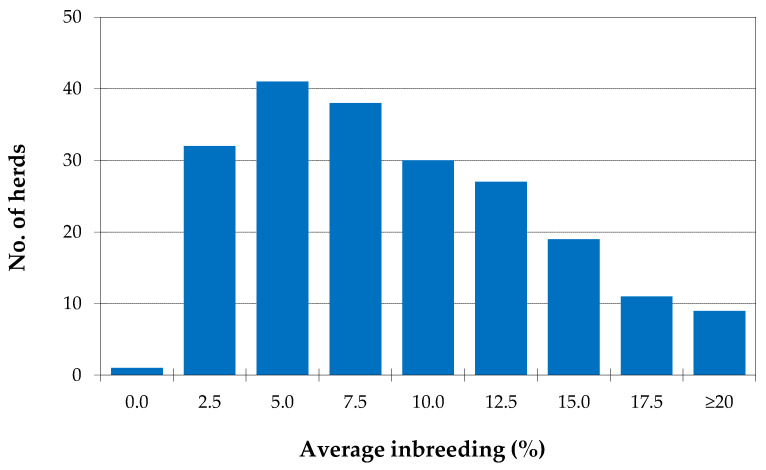
Average inbreeding per herd (only herds with more than 10 calves produced), for calves born from 2015 to 2019.

**Figure 9 animals-10-01990-f009:**
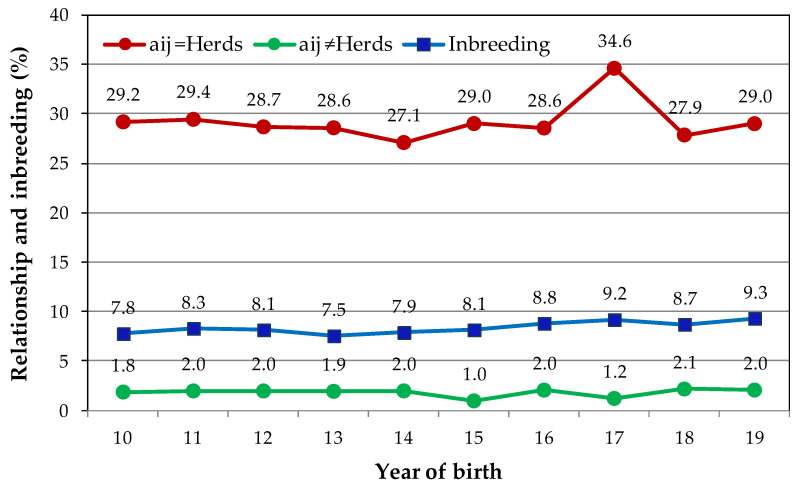
Average relationship (*a_ij_*) for animals born in same herd and born in different herds and inbreeding for calves born from 2010 to 2019.

**Figure 10 animals-10-01990-f010:**
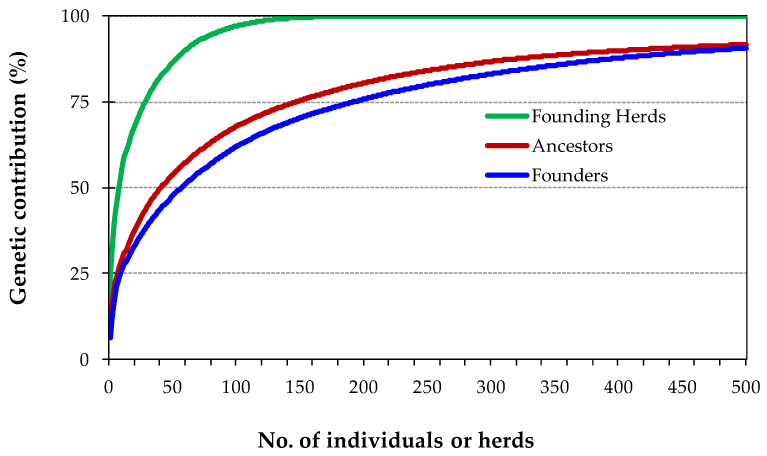
Cumulative genetic contribution to the reference population of the most influential founders, ancestors and herds.

**Figure 11 animals-10-01990-f011:**
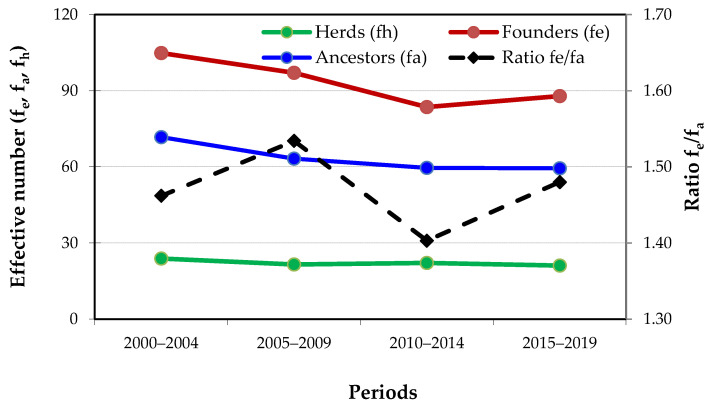
Evolution of effective number of founders, ancestors, herds and ratio *f_e_/f_a_* in 5-year periods.

**Table 1 animals-10-01990-t001:** Mean generation intervals ± standard deviation (SD) for all calves born and for the 4 paths of selection.

Item	Generation Interval, Years		
Sires of calves	6.40 ± 2.61		
Dams of calves	7.09 ± 3.70		
Sires of sires	6.05 ± 2.51	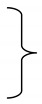	*L* = 7.05 years
Sires of dams	6.07 ± 2.47		
Dams of sires	7.25 ± 3.70		
Dams of dams	8.82 ± 3.63		

**Table 2 animals-10-01990-t002:** Inbreeding and relationship coefficients (±SD) for all animals and for the reference population (calves born between 2015 and 2019, *n* = 35,017).

Item	
All animals (*n* = 209,503)	
Number of generations known	3.98 ± 1.62
Average inbreeding coefficient (%)	7.04 ± 9.46
Animals with inbreeding coefficient ≠ 0 (%)	63.82
Average relatedness (%)	1.76 ± 1.55
Reference population (*n* = 35,017)	
Number of generations known	5.62 ± 1.11
Average inbreeding coefficient (%)	8.82 ± 10.37
Animals with inbreeding coefficient ≠ 0 (%)	86.00
Average relatedness (%)	2.05 ± 1.26
Mean relationship in the reference population (%)	2.23 ± 2.46
Animals from the same herd	29.25 ± 9.36
Animals in different herds	1.87 ± 1.53
ΔF ^1^/year (%)	0.183 ± 0.020
ΔF/generation (%)	1.29
Effective population size ^2^	38.83
Effective population size ^3^	41.69

^1^ ΔF = the rate of inbreeding; ^2^ According to Falconer and MacKay [1]; ^3^ According to Gutierrez et al. [17].

**Table 3 animals-10-01990-t003:** Genetic contributions of founders, ancestors and founding herds to the reference population (calves born between 2015 and 2019, *n* = 35,017).

Item	
Number of founders	11,471
Founder genome equivalents (*ƒ_g_*)	48.7
Effective number	
Founders (*f_e_*)	87.9
Ancestors (*f_a_*)	59.4
Founding herds (*f_h_*)	21.4
Herds supplying sires	73.5
Herds supplying grandsires	33.5
Herds supplying great-grandsires	17.8
Ratio *f_e_*/*f_a_*	1.48
Contribution to 50% of the genetic pool	
Founders	56
Ancestors	41
Founding herds	9
Contribution of 5 most influential founders (%)	19.2
Contribution of 5 most influential ancestors (%)	22.6
Contribution of 5 most influential founding herds (%)	39.6

**Table 4 animals-10-01990-t004:** Summary of demographic parameters reported for different cattle breeds, including generation intervals (*L*), number of generations known (*n_i_*), average relatedness (*AR*), average inbreeding (*F_x_*), percentage of inbred animals (*F_x_* ≠ 0), effective population size (*N_e_*), effective number of founders (*f_e_*) and ancestors (*f_a_*), *f_e_*/*f_a_* ratio and rate of inbreeding per year (ΔF/year).

Breed	Country	*L*	*n_i_*	*AR*	*F_x_*, %	*F_x_* ≠ 0, %	ΔF/year, %	*N_e_*	*f_e_*	*f_a_*	*f_e_*/*f_a_*	Source
Braunvieh	Austria							109	97	52	1.87	[53]
Grauvieh	Austria							72	113	39	2.90	[53]
Pinzgauer	Austria							232	66	47	1.40	[53]
Simmental	Austria							258	221	114	1.94	[53]
Simmental	Brazil	8.0	14.0	0.99	1.49	30.00		48	163	132	1.23	[31,32,33,34,35,36,37,38,39,40,41,42,43,44,45]
Brahman	Brazil	4.4	3.2		11.97	92.00			5	5	1.00	[54]
Bonsmara	Brazil	3.2	2.2	0.81	0.26	8.97		325	220	85	2.59	[45]
Marchigiana	Brazil	7.0	4.5	4.04	1.33	60.43		140	120	32	3.75	[45]
Blanco Orejinegro	Colombia	4.6	6.0	4.60	3.10	100.00			55	38	1.45	[55]
Charolais	Denmark	5.5	8.5		1.04		0.02	558	512	107	4.79	[51]
Limousine	Denmark	5.4	7.3		1.02		0.01	1667	310	92	3.37	[51]
Charolais	France	5.7	9.3		0.67		0.02	493	547	212	2.58	[51]
Limousine	France	6.1	6.9		0.71		0.00	2459	468	156	3.00	[51]
Charolais	Ireland	6.4	9.1		0.99		0.03	244	475	75	6.33	[51]
Limousine	Ireland	6.3	6.5		0.79		0.02	345	395	110	3.59	[51]
Angus	Ireland						0.02		160	40	4.00	[56]
Charolais	Ireland								357	58	6.16	[56]
Hereford	Ireland						0.13	64	150	35	4.29	[56]
Limousin	Ireland								316	82	3.85	[56]
Simmental	Ireland						0.06	127	55	35	1.57	[56]
Calvana	Italy	10.3	10.0	6.39	2.54		0.25	20				[57]
Charolais	Italy	6.7	18.0	0.20	0.55		0.08	90				[57]
Limousine	Italy	7.1	15.0	0.20	0.37		0.05	133				[57]
Mucca Pisana	Italy	8.9	14.0	10.54	2.70		0.30	19				[57]
Pontremolese	Italy	12.5	13.0	7.15	3.42		0.27	15				[57]
Sarda	Italy	10.6	11.0	0.04	3.00		0.28	17				[57]
Sardo Bruna	Italy	13.3	10.0	0.05	2.64		0.20	19				[57]
Sardo Modicana	Italy	7.8	12.0	0.37	1.26		0.16	40				[57]
Chianina	Italy							256	221	96	2.30	[58]
Maremmana	Italy							111	143	120	1.19	[58]
Mucca Pisana	Italy							20	12	12	1.00	[58]
Japanese Brown-Kouchi	Japan	10.4	8.9		8.80			6	79			[52]
Japanese Brown-Kuma.	Japan	9.4	10.2		7.10		0.74	26	74			[52]
Japanese Black	Japan						0.091	17				[59]
Alentejana	Portugal	6.5	3.0		5.74	54	0.33	23	122	55	2.21	[5]
Alentejana	Portugal	6.5	7.5	3.80	9.60	98	0.23	35	84	39	2.17	[38]
Barrosã	Portugal	6.9	4.0	32.90	5.22	53	0.28	26	471	261	1.81	[39]
Preta	Portugal	6.9	3.4	0.90	6.00	59	0.14	53	82	64	1.28	[40]
Jarmelista	Portugal	6.1	2.8	19.00	14.00	90	1.02	9	11	6	1.89	[41]
Marinhoa	Portugal	6.5	4.9	2.34	5.58	90	0.27	29	53	28	1.90	[42]
Arouquesa	Portugal	6.5	4.1	0.44	1.11	38	0.17	45	232	115	2.02	[43]
Cachena	Portugal	6.8	3.0	0.80	6.00	45	0.28	26	208	159	1.31	[44]
Blonde d’Aquitaine	Slovakia		8.0	1.25	0.14			468	136	68	2.00	[60]
Charolais	Slovakia		9.0	0.55	0.47			153	381	139	2.74	[60]
Limousine	Slovakia		8.0	0.62	0.14			429	324	123	2.63	[60]
Simmental	Slovakia		10.0	3.53	1.90			48	58	28	2.07	[60]
Afrikaner	S. Africa	6.6	1.9	0.44	1.83			168	288	226	1.27	[61]
Asturiana de los Valles	Spain	5.4	2.2		1.42	38						[49]
Avilenã Negra Iberica	Spain	5.9	4.0		6.11	81						[49]
Bruna dels Pirineus	Spain	6.6	1.0		1.00	4						[49]
Morucha	Spain	6.4	2.2		5.93	56						[49]
Pirenaica	Spain	6.6	4.6		3.45	95						[49]
Retinta	Spain	6.2	3.8		7.22	73						[49]
Rubia Gallega	Spain	7.1	3.1		2.69	59						[49]
Lidia	Spain	7.5	4.5		7.80			36	28	16	1.75	[50]
Alistana	Spain	4.1	1.5	0.73	1.09	11	0.33	36	265	56	4.73	[48]
Asturiana de la Montaña	Spain	4.6	1.6	0.68	1.55	16	0.31	35	119	83	1.43	[48]
Asturiana de los Valles	Spain	4.3	1.1	0.26	0.48	4	0.13	89	846	163	5.19	[48]
Avilenã Negra Iberica	Spain	3.7	2.2	0.10	2.50	32	0.22	40	68	59	1.15	[48]
Bruna dels Pirineus	Spain	5.5	0.8	0.35	0.25	2	0.09	95	48	40	1.20	[48]
Morucha	Spain	4.8	1.2	0.30	2.20	17	0.36	27	130	105	1.24	[48]
Pirenaica	Spain	6.1	3.0	1.58	1.60	48	0.07	123	153	58	2.64	[48]
Sayaguesa	Spain	3.8	1.7	1.70	3.13	25	0.59	21	116	25	4.64	[48]
Charolais	Sweden	4.6	8.3		0.92		0.00		371	99	3.75	[51]
Limousine	Sweden	5.1	7.5		1.08		–0.02		274	77	3.56	[51]
Limousine	UK	5.9	7.5		1.13		–0.01		232	86	2.70	[51]
Hereford	USA						0.12	85				[61]

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
