# Peer review of "Genetic Diversity in the Portuguese Mertolenga Cattle Breed Assessed by Pedigree Analysis"

_animals, 2020, doi:10.3390/ani10111990_

Round 1

Reviewer 1 Report

Minor comments.-

If the generation interval is between 6,5 and 7 years why the authors select as reference population the animals born the last five years instead of, for instance, the animals born within the las generation?

Figure 7-a. misspelling relatedness? shouldn't be relatedness)

Figure 7-b. Are correct the years in the abscissa axis?

General comments.-

The paper is very well written, making an exhaustive description of the main population genetic parameters estimated using genealogical information. The methodology is correct, and correctly discussed.

The only lower positive comments come from the nature of the limited biological material included, which is reduced to a unique local breed and the nature of the research which is limited to a description of the genetic structure using standard methodologies implemented into a 15 years old software.  

Author Response

The authors gratefully thank to the Referees for the constructive comments and recommendations which definitely help to improve the readability and quality of the paper. All the comments are addressed accordingly and have been incorporated to the revised manuscript.

If the generation interval is between 6,5 and 7 years why the authors select as reference population the animals born the last five years instead of, for instance, the animals born within the las generation?

We appreciate the reviewer's comment but the fact that we chose a reference population of 5 years and not the corresponding to one generation interval (from 6.5 to 7 years on average) was due to the fact that we presented several analyzes by intervals of 5 years time and thus the consistency of the analyzes is maintained. Moreover, the reference population with 5 years of data presents a significant number of animals from the herdbook in relation to the total population under study (35,017 in 209,503; 16,7% of data).

Figure 7-a. misspelling relatedness? shouldn't be relatedness)

Corrected accordingly.

Figure 7-b. Are correct the years in the abscissa axis?

Corrected accordingly.

General comments.-

The paper is very well written, making an exhaustive description of the main population genetic parameters estimated using genealogical information. The methodology is correct, and correctly discussed.

The only lower positive comments come from the nature of the limited biological material included, which is reduced to a unique local breed and the nature of the research which is limited to a description of the genetic structure using standard methodologies implemented into a 15 years old software.

In fact, some of the methods used were proposed several years ago, namely those of James (1972) and Lacy (1989). However, the amount of genealogical information that currently exists, as well as the computational capacity, allows us to estimate parameters based on larger data sets, which was more difficult a few years ago. Likewise, as we get more information about pedigrees, we can do analyzes for different periods over time and, with that, make “x-rays” to the population dynamics and it´s evolution.

Reviewer 2 Report

The manuscript presents a pedigree analysis of the Mertolenga cattle breed in Portugal. Although straightforward in technical terms, the manuscript is well written, easy to follow and presents an interesting case study for the conservation of indigenous cattle breeds. The paper would be thus a very nice addition to the literature. I recommend its publication subject to the minor changes outlined below. 

Minor changes

  1. Figure 9. Why is 2017 such an outlier? It’d be interesting to provide a possible explanation, ie. Change in the recording methods?
  2. L33 and L37. Please check the use of commas and points for numbers through the text.
  3. L292 Include % (inbreeding in the reference population). Check also other parts of the text.

Author Response

The authors gratefully thank to the Referees for the constructive comments and recommendations which definitely help to improve the readability and quality of the paper. All the comments are addressed accordingly and have been incorporated to the revised manuscript.

Figure 9. Why is 2017 such an outlier? It’d be interesting to provide a possible explanation, ie. Change in the recording methods?

The method used was always the same. In fact, the average kinship in 2017 is higher, and there is not a single obvious reason to this value. Several factors may occasionally contribute to the increase of the average kinship on the herd, as for economic reasons the breeders acquired less males in other herds.

L33 and L37. Please check the use of commas and points for numbers through the text.

Corrected accordingly.

L292 Include % (inbreeding in the reference population). Check also other parts of the text.

Corrected accordingly.

Reviewer 3 Report

The manuscript evalueted the population structure and genetic diversity of the Mertolenga breed. The results obtained are important for monitoring and assisting decisions about the sustainability of the breed. I understand that the manuscript makes an excellent contribution to the genetic management of the breed. Therefore, i recommend being accepted for publication.

Author Response

We appreciate the positive comments.

Reviewer 4 Report

I would like to congratulate the authors for the excellent research. The article is very well written, uses appropriate methods, presents consistent results that support the conclusions and deals in detail with an important local genetic resource (Portuguese Mertolenga cattle). My few considerations are in the sense of enriching the material.

L97. Define MHB

Figure 2. Change 'Cows Pure Line' to 'Purebred cows'

Figure 2. The reader understands that the calves shown in the figure 2 are not necessarily purebred. The number of purebred calves could have been indicated here.

Figure 3 and L219-220. It would be interesting to build the same figure (side by side with the current one) restricted to animals born within the last generation interval (last ~ 7 years or 2015-2019). Probably some impact of the breeding program on the use of breeding animals would be evidenced.

Figure 7a. Was the red font on the second vertical axis intentional? It would be more interesting to show generation equivalents than known generations

Table 2. It would be interesting to have calculated the effective population size from the rate of coancestry, especially since the present population probably has a subdivision. See Cervantes et al. (2011) doi:10.1111/j.1439-0388.2010.00881.x. I believe that Endog can provide you with this parameter or you can calculate it by randomly sampling individuals from the population [Leroy et al. (2013) doi: 10.1186 / 1297-9686-45-1]. The inclusion of this parameter could enrich the discussion of your study (L477-489).

The 'Discussion' section is fragmented into some short paragraphs (<6 lines). It would be appropriate not to separate the same subject into paragraphs. Example: L386-392, 402-404, 430-449, 450-464, 477-489...

L402-404. I found the data presented in Table 4 to be important, however, this does not seem to me a review study. Thus, I believe that this material could be included as a supplement.

L532. I consider such a large amount of references to be an exaggeration to justify a certain comment. This has been verified in some parts of your discussion. Usually, 3 references are sufficient. Also check the rules for this journal.

L534. Conclusions. I suggest drastically reducing your conclusions. Tip: Align your conclusions with the title and objective of the study. Be more objective. Much of the text included here could be considered to enrich the 'Discussion' section. 

Author Response

The authors gratefully thank to the Referees for the constructive comments and recommendations which definitely help to improve the readability and quality of the paper. All the comments are addressed accordingly and have been incorporated to the revised manuscript.

L97. Define MHB

Corrected accordingly.

Figure 2. Change 'Cows Pure Line' to 'Purebred cows'

Corrected accordingly.

Figure 2. The reader understands that the calves shown in the figure 2 are not necessarily purebred. The number of purebred calves could have been indicated here.

Corrected accordingly.

Figure 3 and L219-220. It would be interesting to build the same figure (side by side with the current one) restricted to animals born within the last generation interval (last ~ 7 years or 2015-2019). Probably some impact of the breeding program on the use of breeding animals would be evidenced.

We cannot understand the reviewer comments. Figure 3 shows the number of active breeding females and calves registered in the Mertolenga’s Herdbook by year and coat variety.

Figure 7a. Was the red font on the second vertical axis intentional? It would be more interesting to show generation equivalents than known generations

Corrected accordingly.

Table 2. It would be interesting to have calculated the effective population size from the rate of coancestry, especially since the present population probably has a subdivision. See Cervantes et al. (2011) doi:10.1111/j.1439-0388.2010.00881.x. I believe that Endog can provide you with this parameter or you can calculate it by randomly sampling individuals from the population [Leroy et al. (2013) doi: 10.1186 / 1297-9686-45-1]. The inclusion of this parameter could enrich the discussion of your study (L477-489).

The reviewer is correct making that suggestion and we really thought about including several methodologies to estimate the Ne. However, due to the fact that in recent years the births of animals without known parents have been recorded, as indicated in lines L242 (results section) and L412 (discussion section), there has been a decrease in coancestry for the reference population (animals born in 2015- 2019) which would cause an estimate of the Ne quite biased and incongruous when compared to the other estimates. We agree with the reviewer that it is important to estimate some indicators in different ways, but we would have to ensure that they are consistent between them.

The 'Discussion' section is fragmented into some short paragraphs (<6 lines). It would be appropriate not to separate the same subject into paragraphs. Example: L386-392, 402-404, 430-449, 450-464, 477-489...

Corrected accordingly.

L402-404. I found the data presented in Table 4 to be important, however, this does not seem to me a review study. Thus, I believe that this material could be included as a supplement.

The inclusion of Table 4 in the body of the text was intentional to facilitate the comparison of results and to support our discussion. The reviewer's comment is pertinent, but we would prefer to keep the table in the text instead of sending it as an attachment that would be much less visible for the readers.

L532. I consider such a large amount of references to be an exaggeration to justify a certain comment. This has been verified in some parts of your discussion. Usually, 3 references are sufficient. Also check the rules for this journal.

The journal does not mention the limit of references and, in some cases, we understand that  citing several authors values the discussion.

L534. Conclusions. I suggest drastically reducing your conclusions. Tip: Align your conclusions with the title and objective of the study. Be more objective. Much of the text included here could be considered to enrich the 'Discussion' section. 

Corrected accordingly.

As the reviewer suggested, we have shorten the conclusions, passing a paragraph to the discussion of the results chapter and have reduced the total number of paragraphs in this chapter.